# Filler Migration after Facial Injection—A Narrative Review

**Uwe Wollina** [1],* and **Alberto Goldman** [2]

1   Department of Dermatology and Allergology, Städtisches Klinikum Dresden, Academic Teaching Hospital, 01067 Dresden, Germany
2   Department of Plastic Surgery, Hospital São Lucas da PUCRS, Porto Alegre 90610-001, Brazil; alberto@goldman.com.br
*   Correspondence: uwollina@gmail.com

**Abstract:** Background: The injection of dermal fillers for facial esthetics has become a very popular procedure. Although usually safe in the hands of the experienced user, filler injections may bear a risk of unwanted side effects. Material and Methods: This is a narrative review of dermal filler migration after facial injections. We performed research on the literature on Pubmed and Google Scholar. Inclusion criteria were observational studies, case reports, and clinical trials which investigated the association of facial filler injections to filler migration. Animal studies have not been considered. Intravascular injections were excluded. Results: We identified 28 reports that met the inclusion criteria. The age range of affected patients was 21 to 86 years (mean ± standard deviation: 47 ± 14.8 years). Women were 25 times more reported than males. Hyaluronic acid and polyalkylimide were the most commonly encountered filler substances. Injections into the nose, lips, nasolabial folds, and forehead (including glabella) are more often reported for filler migration than injections into the cheeks. Tear-trough correction bears a risk for orbital migration. The delay from injection to presentation of filler migration was highly variable. Very late filler migration was more commonly seen with permanent fillers than non-permanent products. Conclusions: Filler migration distant from the injection site can occur even several years after the primary treatment. All filler types can be involved. Permanent fillers bear a higher risk of very late filler migration. Migration of permanent fillers needs surgical treatment, while HA fillers respond to hyaluronidase injections. Detailed knowledge of facial anatomy, safer injection techniques, and filler qualities are preventive measures.

**Keywords:** dermal fillers; facial esthetics; filler migration; adverse events

## 1. Introduction

The use of fillers in esthetic medicine has grown faster than any other procedure, reflecting a trend for minimally invasive procedures. The fillers represent versatile tools in facial rejuvenation, contouring, and volumizing. They can be classified according to substance(s), biophysical characteristics, crosslinking methods, and longevity of lifting effect [1–3].

Temporary fillers include collagen and hyaluronic acid (HA)-based products. Semipermanent (also known as biostimulatory) fillers are calcium hydroxyapatite (CaH), polycaprolactone, and poly-D, L-lactic acid (PDLLA). Polymethyl-methacrylate (PMMA), polyalkylimide, polyacrylamide, paraffin, polydimethylsiloxane (PDMS), and other silicones are the substances used as permanent fillers. Paraffin was one of the first substances used as filler, but safety issues and dissatisfying long-term outcomes made it disappear in medical esthetics. On the contrary, silicones have never been approved for use as filler, but sometimes they are still used off-label [2].

There is no ideal filler. The adverse events can be classified into acute or immediate (during and shortly after injection), early (up to 2 weeks after injection), delayed (>2 weeks to 12 months), late (>1 year), and localized or generalized [4]. The most common adverse events of the immediate or early type of reaction are injection pain, bruising, and temporary

redness, while nodules and granulomas are usually delayed or late reactions. Vascular compromise is an acute and severe adverse event caused by intraarterial or intravenous injections, extraluminal pressure of filler material, or indirect vascular trauma. Fortunately, both abscess formation and vascular occlusion are much less frequent [5,6]. The rate of adverse events is widely dependent on the skills and experience of the user. Another problem is the participation of laypersons without medical understanding in illicit filler injections [7–9]. The general complication rate of filler injections in experienced hands is less than 5% [10]. A summary of possible adverse events after facial filler injection is given in Table 1 [11–33].

**Table 1.** Possible adverse events associated with facial filler injection (Ref.: reference(s)).

| Adverse Events | Remarks | Ref. |
| --- | --- | --- |
| Abscess | Needs microbial confirmation and identification of responsible microorganisms, treatment mostly surgical | [11] |
| "Angry red bump" | Indurated painful, petrified nodule or plaque, may occur within the first month after injection or later | [12] |
| Alopecia, non-scarring | After temporal injection | [13] |
| Acute diplopia | Vascular compromise after temporal injection, severe pain | [14] |
| Bruising | More often around the eyes | [15] |
| Cerebral infarction and stroke | Most common after accidental injection in the ophthalmic angiosome | [16,17] |
| Delayed-onset nodules | Various pathologies for inflammatory to infectious or filler migration | [18] |
| Discoloration | More often seen around the eyes, sometimes due to Tyndall effect, sometimes due to post-inflammatory pigmentary changes | [19] |
| Dizziness and pain | Vascular compromise after temporal injection, severe pain | [20] |
| Edema, recurrent | Inflammatory granulomatous reaction | [21] |
| Erythema | Mostly temporary, common | [15] |
| Filler embolism | Risk depending on anatomical area, filler volume, and type | [22] |
| Foreign body granuloma | Needs histologic confirmation, may be delayed for years, can be due to filler migration | [23] |
| HA—hypersensitivity | Seen occasionally after SARS-CoV-2 infection | [24] |
| Inflammatory granuloma | Needs histologic confirmation, may be delayed for years, can be due to filler migration | [25] |
| Infectious granuloma | Needs histologic and microbial confirmation | [26] |
| Lipogranuloma | Needs histologic confirmation, can be delayed for years | [27] |
| Sarcoidosis | Reported after polycaprolactone injection | [28] |
| Soft tissue necrosis | Vascular compromise by embolization or extravascular pressure on blood vessels | [29] |
| Soft tissue infection and sepsis | Unmeet hygienic standards, contaminated filler material | [30] |
| Tongue necrosis | After injection into the chin, vascular compromise | [31] |
| Visual loss | Most often after injection into glabella or nose with acute ocular pain, lid ptosis, and ophthalmoplegia due to vascular compromise | [32] |
| Vitiligo | Reported after polycaprolactone injection | [33] |
| Xanthelasma | Mostly seen on eyelids after multiple injections over time | [34] |

While HA fillers may be removed by injecting the enzyme hyaluronidase, permanent filler reactions often warrant surgery for complete removal. The kinetics of HA degradation by hyaluronidase is dependent on crosslinking filler technology, but not the concentration of HA [35]. Permanent fillers have been associated with possible delayed adverse events occurring even years after injection and with unwanted systemic side effects including myalgia, arthralgia, fever, lymphadenopathy, and malaise [36,37]. A great danger for patients arises from the use of illegal injectable fillers [38].

To ensure patient safety and avoid mistakes, various guidelines have been developed [39–41].

In this narrative review, we will focus on a lesser-known and rarer adverse event after facial filler injections, i.e., the delayed filler migration or displacement, often observed weeks to years after injection. Filler migration is defined as filler material found remote from

the original site of injection. To narrow this review, we excluded inadvertently intravascular injections that might also cause filler migration as an acute severe adverse event.

## 2. Materials and Methods

We performed research on the literature through Pubmed and Google Scholar. Inclusion criteria were observational studies, case reports, and clinical trials which investigated the association of facial filler injections to filler migration. Animal studies have not been considered. The search string, designed to cover the primary themes of the review and optimized for high sensitivity, was (dermal fillers [Title/Abstract] OR AND (facial esthetics [Title/Abstract] AND migration [Title/Abstract] OR filler migration [Title/Abstract]. Exclusion criteria were intravascular injection, acute displacement, vascular compromise, and infection. The identified articles were included for further analysis if the title or the abstract showed the matching keywords. We decided not to specify a year range for the selection of articles. The articles were excluded if they contained only abstracts, and were not in the English or German language. Papers published in conferences, books, or book chapters were also excluded. We also searched further from the references of the articles which showed our area of concern (Figure 1).

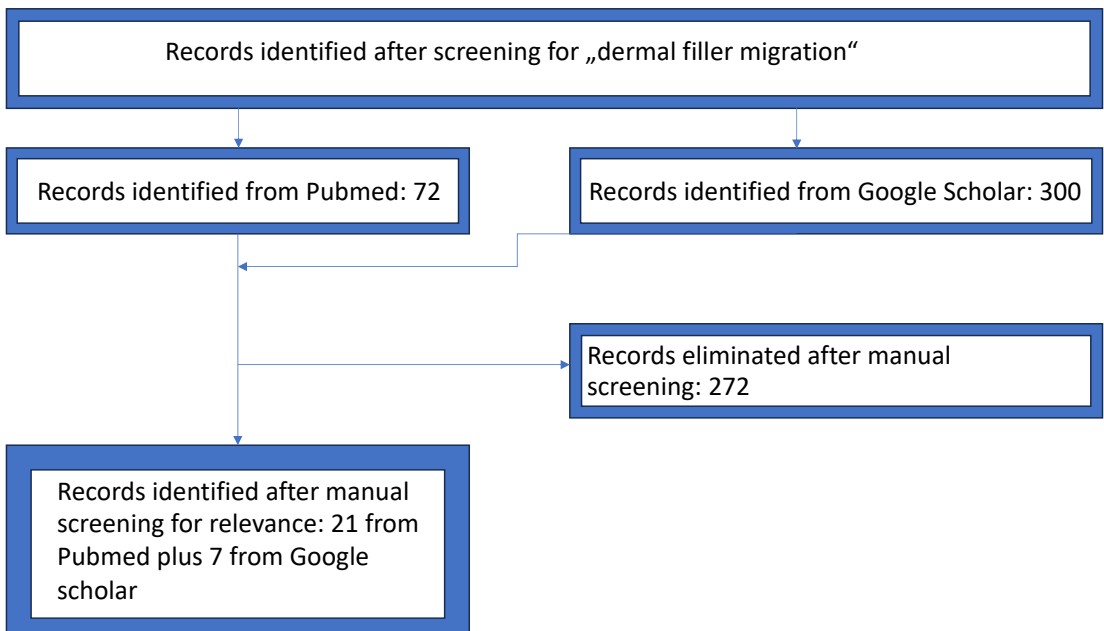

**Figure 1.** PRISMA selection flowchart.

Filler migration was defined as the presence of filler at a location distant from the injection site without accidental intravascular injection.

## 3. Results

Initially, we identified 43 papers. After the first check, 22 papers were excluded according to the exclusion criteria. Analyzing these papers, 7 additional reports were found [42–70]. A summary is provided in Table 2.

The age range was 21 to 86 years (mean ± standard deviation: 47 ± 14.8 years). Women were 25 times more reported than males. HA and polyalkylimide were the most commonly encountered filler substances. CaH, PDLLA, PDMS, and other silicones, and PMMA and polyacrylamide were identified in selected cases. Injections into the nose, lips, nasolabial folds, and forehead (including glabella) seem to be more often reported for filler migration than injections into the cheeks. While nasolabial folds and lips are major targets for filler placement, the nose and forehead are less commonly treated. Muscular activity is highest on the lips followed by glabella. On the cheeks, deep injections are favored for

facial contouring in contrast to the other areas mentioned. Deposition of fillers into deep midfacial fat pads bears obviously a lower risk for filler migration. Tear trough correction carries a risk for orbital migration.

**Table 2.** Reported cases of filler migration after facial esthetic procedures (Ref.: reference).

| Ref. | Patient | Filler | Clinical Findings | Treatment |
|---|---|---|---|---|
| **Forehead and glabella** | | | | |
| [64] | 52-y; f | Polylakylimide, 10 y after injection into the glabella | Swelling of left brow, temple, and glabella | Surgery |
| **Tempel** | | | | |
| [46] | 34-y, f | HA, 2 weeks after temporal injection | Bipolar buccal masses | Hyaluronidase |
| **Nose** | | | | |
| [42] | 71-y, f | HA, 10 months ago, right in the nasal radix | 15 mm tumor on the dorsolateral nasal wall extrusion | Needle puncture and |
| [54] | 33-y, f | HA, 16 y after injection into the nose | Bean-sized mass on the forehead (2×) | Surgery |
| [67] | 37-y, f | CaH, 3 days after injection into the nose | 6 × 2 cm measuring mass of the left eyebrow and upper eyelid | Surgery |
| **Periocular region** | | | | |
| [50] | 63-y, f | HA, 1 year after Injection into forehead and lateral eyebrow | Palpable masses in the right anterior superior orbit, mild lid swelling | Surgical excision |
| [53] | 54-y, f | Unknown filler, 7 y after injection in the lower eyelids | Right upper eyelid nodule | Surgery |
| [62] | 48–56-y, 3 f | HA, 6 months to 5 y after correction of tear trough | Bluish nodules with a Tyndall effect on lower eyelids | Surgery |
| **Cheeks** | | | | |
| [44] | 41-y, f | HA, 2 weeks ago, on zygomatic protrusion | Subconjunctival nodule on the left eye | Incision and extrusion |
| [47] | 57-y, f | HA, 3 y after injection In the zygomatic area | Right lower eyelid mass | Surgery |
| **Nasolabial fold** | | | | |
| [61] | 48–71-y, 5 f | CaH, 2–12 months after injection into nasolabial folds and perioral region | Intraoral nodules (upper lip mucosa and vestibulum oris) | Unspecified |
| [63] | 86-y, m | Polyalkylimide, several months ago into the nasolabial fold | Plaques of the left lower eyelid | Biopsy |
| | 46-y, 48-y, f | HA, after 1 and 6 y after injection into the nasolabial fold | Swelling along the lower orbital rim, swelling of the right lower eyelid | Biopsy |
| [65] | 52-y, f | CaH, 1 y after injection into nasolabial fold | Inflammatory nodule of vermillion border of the right upper lip | Biopsy |
| **Lips** | | | | |
| [43] | 34-y, f | HA, 3 months after lip augmentation | 2 cm mobile mass of the right cheek | Surgery |
| [51] | 35-y, f | PDMS, 9 months after injection into the lip | Massive edema of face and neck, inflammation, nodular irregularities | Corticosteroids with limited effect, surgery |
| [57] | 28–74 y, 26 f | Silicone, HA, PMMA, polyalkylimide, Polyacrylamide and others; few months to 8 years after injection into the lips | The number of patients with migration only is unknown | Corticosteroids, antibiotics, aspiration, surgery |

**Table 2.** *Cont.*

| Ref. | Patient | Filler | Clinical Findings | Treatment |
|---|---|---|---|---|
| Multiple localizations | | | | |
| [45] | 42–67 y, 6 f, 1 m | HA, up to 10 y after injection into cheeks, naso-labial folds or forehead | Eyelid swelling, lid mass, nasolacrimal duct obstruction or neurologic deficit | Surgery (6×), hyaluronidase (1×) |
| [48] | 52-y, f | HA, 2 y ago, on lips and naso-labial folds | Buccal "tumor", multiple granulomas with inflammation | Surgical excision |
| [49] | 24–52-y, 16 × f | Polyalkylimide, 3–7 y after injection in the nasal bridge, temporal region or cheeks | Lower eyelid swelling | Surgical excision, 1 × relapse, 1 × lid retraction |
| [52] | 50-y, f | HA and PDLLA, obsessive patient after multiple middle and lower face treatments, 1 y after injections into the cheeks | Palpable mass from the forehead to the nose | Biopsy, surgery suggested |
| [55] | 44-y, 57y, 77-y; f | HA, 2 × after HA injection into the nasolabial fold 1.5 and 4 y ago, 1 × HA + acrylamide particles for temporal augmentation 10 y ago | 2 × eyelid swelling, 1 × non-inflammatory nodule of the left lateral canthus | Hyaluronidase for HA, surgery offered for HA +acrylamide |
| [66] | 45-y, 48-y, 51-y, f  62-y, m | Polyalkylimide, 10 to 245 months after injection into cheeks, nasolabial folds or tear trough | Irregularities and edema of periorbital area | Aspiration, surgery |
| [68] | 59-y, f | CaH, 2 weeks after injection into naso-labial folds and marionette lines | Nodule of the right vermillion | Biopsy |
| * | 40-y, 55-y, 64-y, f  33-y, m | PDLLA, PMMA, silicon, injection in lower and midface, glabella, or mouth commissure between 4 months to 12 y ago  PMMA, injection into glabella, several months ago | Subcutaneous nodules submandibular and on the neck, nodules on the nasal radix and dorsum, nodules in the mentolabial fold  Subcutaneous nodules on nasal radix | Surgery (plus intralesional Nd-YAG laser)  Intralesional Nd-YAG laser |
| Unknown specific injection site(s) | | | | |
| [56] | 60.8-y (mean), 57 f | Permanent fillers, mean follow-up 16.6 y | Migration into cervical lymph nodes in 59.6% | Not mentioned |
| [58] | 55-y, f | Silicone, after multiple facial injections 50 to 60 y ago | 6 × 8 cm large pseudocyst of the neck | Surgery, antibiotics |
| [59] | 74-y, f | Filler not identified, facial injection 3 y ago | Nodule of the left upper eyelid probably silicone | Biopsy only |
| [60] | 25–76 y, 16 f, 16 m | PMMA, polyacryl-amide, polyalkylimide, facial injection 6–120 months ago | 5 patients had clinical migration, by MRI 28% (30 of 107) lesions | Surgery |
| [69] | 31–55 y; 18 f | Polyalkylimide, between 1 month to 3 y after facial injection | Irregularities and edema in different areas | Surgery |

Legends: y—years; f—female; m—male. * This publication.

The delay from injection to presentation of filler migration was highly variable, from 2 weeks to 60 years. The extremely late presentation of filler migration was associated with the use of permanent fillers like silicones, polyalkylimide, PMMA, and polyacrylamide.

However, there was a single case report with filler migration presenting 16 years after injection of HA [55].

The clinical presentation varied from nodules to tumor-like masses, some with inflammation, edema, and pain. The possible differential diagnoses ranged from granulomatous to inflammatory disorders, soft tissue infection, cysts, and tumors. Imaging by diagnostic ultrasound or MRI had been reported in several cases. Confirmation of diagnosis was done by histopathology. MRI revealed significantly more lesions due to filler migration than clinically suspected [61]. In an ultrasound study on 57 patients after facial implantation of a permanent filler, lymphatic spread migration to cervical lymph nodes was observed in 34 patients (59.6%) [57].

Treatment was not reported for all patients. Before the final diagnosis was confirmed, corticosteroids and antibiotics have often been prescribed. They were of limited success. In the case of HA, hyaluronidase injections were employed, but in some patients, surgery had been performed since the diagnosis was unsure. For permanent fillers, surgery was the treatment of choice.

Hyaluronidase works faster on HA filler when applied early, and in late reactions, multiple and higher dosages might work well. For semipermanent fillers no specific drug therapy is available. Hydrolysis will eventually decompose PLLA but it is a slow process. In some cases, CaH may need minor surgery, e.g., on the vermillion border, but the injection in the lips is no longer recommended. In the case of most permanent fillers, the particulate material is embedded in a scaffold of collagen fibers which becomes tighter with time.

In addition, we added four patients to our own practice, 3 women and 1 man. Three developed subcutaneous nodules after permanent fillers (silicon and PMMA), and one female presented with subcutaneous nodules after PDLLA injection by a beautician (Figures 1–4).

Due to the variability in clinical presentation, the range of possible differential diagnoses is broad. They include soft tissue infections including abscess and cellulite (erysipelas), atypical mycobacterial infections, herpes zoster, and mycetoma [4,12,15,18]. Non-infectious differential diagnoses include sarcoidosis, scars, xanthelasma, temporal arteritis, neurogenic ulcer, and keratoacanthoma [16,25,28]. Benign and malignant tumors must be considered in other cases such as parotid pleomorphic adenoma, cutaneous cysts, facial granuloma, and pseudo-lymphoma on the benign spectrum, and basal cell carcinoma, Merkel cell carcinoma, cutaneous lymphoma or metastasis as malignancies [42,71–73].

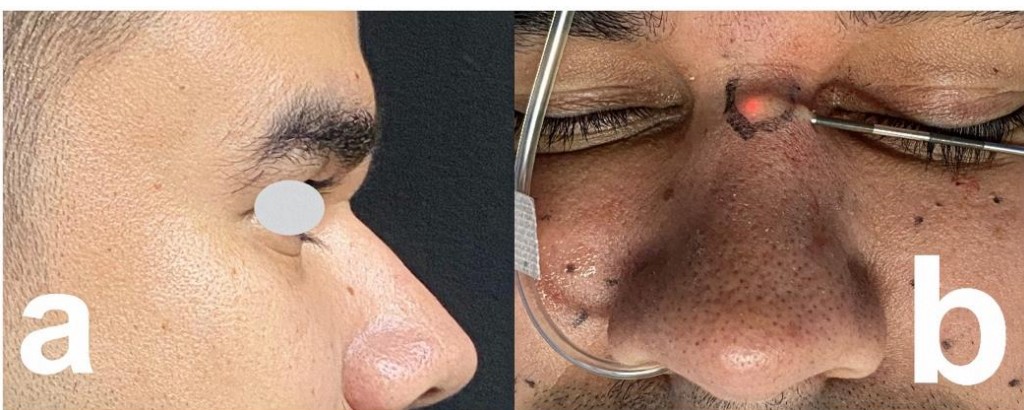

**Figure 2.** A 33-year-old man. (**a**) PMMA injected in glabella with displacement to radix. (**b**) Intralesional laser treatment.

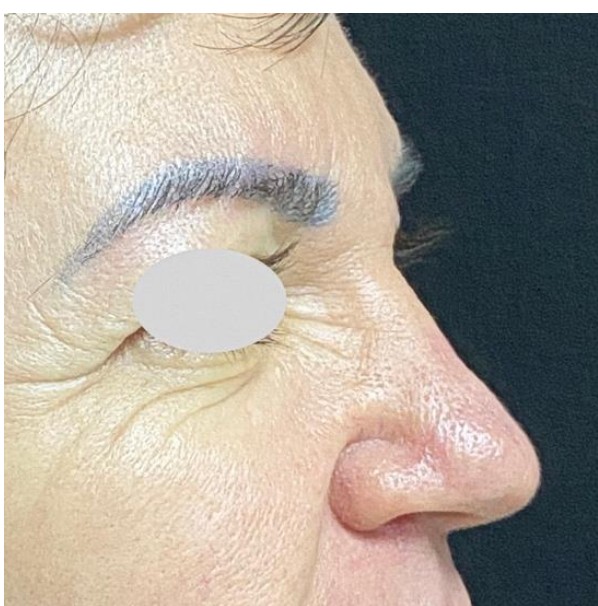

**Figure 3.** 55-year-old woman. Silicone injected 12 years ago into the glabella by an esthetician. Displacement to radix and nasal dorsum.

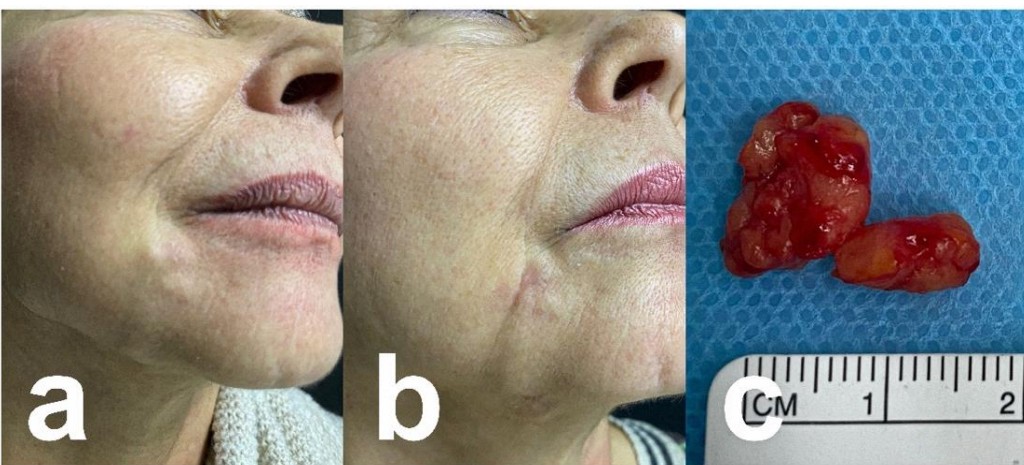

**Figure 4.** A 64-year-old woman. (**a**) PMMA migration (originally injected 20 years ago in mouth commissure) with subcutaneous nodules. (**b**) After surgical excision. (**c**) Surgical specimen.

## 4. Discussion

Amongst possible adverse events of dermal filler injections, migration of filler material is a relatively rare and delayed adverse event [54–58,61]. Considering polyacrylamide filler, migration has been observed in up to 3% of cases [67]. Clinical evidence of filler migration may not be observed unless a secondary complication, such as granuloma, edema, or cysts, developed.

Migration of filler material may result in an unsightly appearance, chronic inflammation, infection, and vascular compromise with disastrous complications including stroke and blindness. Filler migration can be induced by several mechanisms. Common but avoidable is poor injection technique due to lacking skills and poor knowledge of anatomy. Further technical errors are high-volume injection, too-fast injection under pressure, and/or the use of a too-small needle or cannula. Other factors contributing to migration include pressure-induced displacement through massage, muscle activity, gravity, and spreading through the lymphatic system (migrated granulomas) [56]. A neuroradiological study suggested that the risk of migration of filler material tends to be lower with the injection of smaller droplets [74].

The formation of foreign body granulomas is dependent upon macrophage function. The major subtypes are M1 with pro-inflammatory activity and pro-fibrotic M2. An in vitro study using different preparations of a commercially available HA filler, experimental HA filler material, and permanent fillers, and Aquamid® and Bio-Alcamid® investigated gene expression patterns after incubation with human TPH-1 monocytes [75]. M0 macrophages responded with a small increase in interleukin (IL)-1β and chemokine CCL18 to HA, but a strong IL-1β and CCL18 response to both non-resorbable fillers. In contrast to HA, Bio-Alcamid® also stimulated further pro-inflammatory cytokines such as IL-3, IL-4, interferon (IFN) type 1, and macrophage markers [76]. In a second trial, both HA fillers and Bio-Alcamid® were injected into a 3D human Phenion® Full-Thickness Skin model. Both resorbable and non-resorbable fillers induced an up-regulation of cytokines and chemokines, suggesting an acute inflammatory tissue response. Details, however, were different between HA and permanent filler. Bio-Alcamid® increased significant tumor necrosis factor (TNF) α, IL-1β, IL-6, and IL-8, whereas the response to HA fillers was markedly less pronounced [77]. As a rule, particulated fillers bear a higher risk of granuloma formation compared to non-particulated products [78].

Most granulomas develop around injection sites. In these cases, migration is not involved. However, granuloma formation distant from the primary injection is a consequence of the migration of filler material. In the case of permanent fillers, the degradation of larger particles allows phagocytosis by macrophages [79]. Transendothelial migration of macrophages has been demonstrated using near-infrared techniques such as fluorescence reflectance imaging and fluorescence-mediated tomography in experimentally induced permanent filler granulomas (polyacrylamide) [80]. With indocyanine green-labeled monocytes and near-infrared laser angiography, infectious and non-infectious inflammation could be differentiated in an animal model [81].

Aesthetic facial subunits reveal a different risk of filler migration due to variations in static and dynamic anatomy [82]. In addition, the injection plane in facial tissue is important for the longevity of the volumizing effect after filler placement. Injections into midfacial deep fat and lateral temporal-cheek superficial fat compartments provide better longevity compared to the perioral region and chin, in reverse suggesting a lower risk of filler migration in general [83].

Intravascular injections have not been considered in this review since they are acute adverse events in contrast to the other mechanisms of filler migration [15,25]. Here, we present suggested mechanisms of migration or displacement in special areas, corresponding to anatomy:

Injections in the glabella: In the case of strong glabella lines, the nasal bridge is particularly subject to the activity of corrugator supercilii and procerus muscles. Repeated muscle contractions may become responsible for the migration of filler injected in the glabella or radix [42]. Examples are shown in Figures 2 and 3.

Injections into the forehead: Potential mechanisms of filler displacement include migration of the filler through the galea aponeurosis and orbital septum by gravity and facial muscle movement. Fillers injected into the forehead may migrate along the tissue plane formed by the galea and the posterior orbicularis fascia functions to the upper eyelid. Facial muscle movement and gravity are involved. In addition, postoperative massage of the injection sites has also been considered, but this would account for migration in a closer timeframe after the injection [60].

Injections in the temple: The superior boundary of the buccal space is connected to the deep temporal fat pad. The anterior buccal extension inserts into the cheeks. With increasing age buccal extension may develop a pseudo-prolapse into the buccal space of the anterior cheek which can facilitate filler migration from the temple to the cheeks [47].

Injections close to the eye: Injection of fillers in the vicinity of the eye can facilitate their migration under the conjunctiva, most probably due to lymphatic spread [44].

Injection for tear trough augmentation: The tear trough is the landmark of the medial border of sub-orbicularis oculi fat. Tear trough deformities are located between the palpe-

bral and orbital parts of the orbicularis oculi. With age, the orbicularis retaining ligament separating deep and superficial fat pads gets weaker, which may lead to herniation of orbital fat. Injections above the deep plane and too close to the orbital rim can cause unwanted filler migration into the lower eyelid [84]. In a systematic review, the frequency of filler migration in this area was 7.7% [45].

Injections into nasolabial folds: In contrast to other facial regions subcutaneous fat is dispersed. The muscles of facial expression form a strong network of collagenous fibers with a direct connection to the skin. Filler injected into the nasolabial fold may migrate due to gravity and muscular movement into the labiomental sulcus. Displacement of filler material from nasolabial folds has been reported in 0.5% of cases [85,86]. An example is shown in Figure 4.

There are rare reports of filler migration after distant injection for gluteal enhancement liquid silicone or PMMA [51,87,88]. Distant migration of polyacrylamide has been observed after breast augmentation to the axilla, chest wall, abdominal wall, and peritoneum [89]. Some aspects must be considered in these cases: (a) much larger volumes were applied compared to facial esthetics, and (b) PMMA, silicone, and polymethacrylate are permanent fillers. (c) Silicones are used off-label since they have never been FDA-approved for esthetics. It is lesser known that even HA can persist in subcutaneous adipose tissue for several years, which explains very rare documented cases of late filler migration after injection of HA [55,90].

Nonetheless, filler migration or displacement has been reported with both temporary and permanent fillers. The differential diagnosis of the resulting clinical symptomatology is broad and includes infectious, inflammatory, cystic, and neoplastic disorders. Often, patients are not aware of the filler products that have been used. Sometimes, different products were applied simultaneously. It is important to specify the filler material by imaging techniques and/or histopathology. While HA-induced lesions may be treated with intralesional hyaluronidase injections, permanent fillers often need surgery [91]. Occasionally, an allergy to hyaluronidase has been observed. One risk factor is a previous sensitization to bee or wasp venom [92].

We have recently described a combined technique of neodymium (Nd)-YAG laser application followed by minor surgery to remove PMMA particles avoiding extensive surgery, even in such delicate localizations such as the nose and the lips (Figures 2–5) [93,94]. The laser energy is released intralesional using a 300–600 μm blunt fiber embedded in a micro-cannula. The skin temperature is controlled during the procedure to prevent burning. The laser energy causes fragmentation of PMMA particles, which can subsequently be removed by a blunt suction cannula with negative pressure. Satisfaction with this procedure was high.

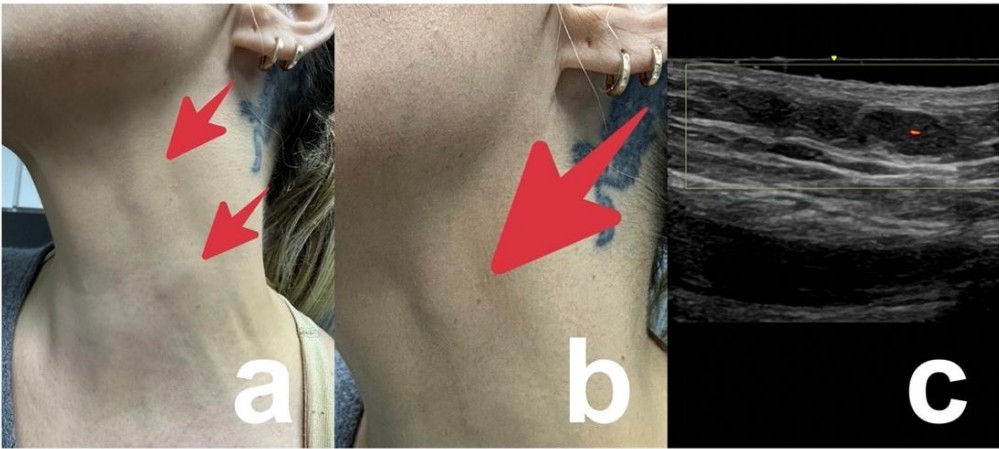

**Figure 5.** 40-year-old women with subcutaneous nodules on the neck 4 months after injections of PDLLA into middle and lower face by a beautician (red arrows). Intralesional corticosteroids applied by the beautician did not improve the nodules but caused localized skin atrophy. (**a**) Overview. (**b**) Detail. (**c**) Doppler ultrasound showing the subcutaneous nodule but excluding lymphadenopathy.

In summary, dermal filler material may spread to distant body sites from the point of injection by vascular transport (blood vessels and lymphatic vessels), along anatomical structures like ligaments and fasciae, by gravity or muscular activity, and along with inflammatory reactions through intracellular transport by macrophages.

Diagnosis of filler migration depends upon medical history and careful clinical examination. Due to the variability in the clinical presentation, numerous differential diagnoses must be excluded. Localization of filler material in deeper tissue layers is supported by imaging techniques. High-frequency ultrasound has been used to localize and identify dermal fillers in the face [95,96]. For HA hypoechoic or anechoic structures are characteristic, while CaH, polycaprolactone, silicon oil, and PMMA demonstrate hyperechoic deposits [97].

Magnetic resonance imaging (MRI) allows not only the localization of filler material but supports the identification of the type of filler material responsible. HA filler show hyperintense signals on T2 W sequences and hypointense signals on T1 W sequences because of their water binding capacity. In contrast, CaH shows low to intermediate signals, and PDLLA T2 W images are hypointense [98].

For final confirmation of a diagnosis of filler migration, histopathology is essential. Since biopsy is an invasive technique, patients sometimes refuse. However, in the case of placement of various or unknown filler materials, it remains the gold standard in diagnostics [5,97,99].

## 5. Conclusions

Filler migration is a less common adverse event seen after injectable filler placement. It can occur on every body part. We focused in this review on facial filler treatment since this is the most common site of application. The migration often occurs with a delay of weeks to years after implantation and is not restricted to certain filler products. However, the management of filler migration differs between HA and permanent fillers. While HA can be removed by intralesional injection of hyaluronidase, permanent filler removal needs surgery. The invasiveness of such a procedure can be reduced by prior intralesional application of the Nd-YAG laser [93,94]. Since the clinical symptoms of filler migration are not uniform, a broad range of differential diagnoses must be considered.

For patient safety, qualified knowledge of anatomy, dermal filler materials, and injection techniques are most important. For beginners, we recommend attending qualified courses by established, serious providers to increase their knowledge and skills. Early recognition and management of unwanted side effects are of great importance to minimize risks of an unfortunate outcome [7,39–41].

**Author Contributions:** Conceptualization, visualization, and writing—original draft preparation, A.G. and U.W.; methodology, investigation, and data procession, A.G. and U.W.; supervision and writing—review and editing, A.G. and U.W. All authors have read and agreed to the published version of the manuscript.

**Funding:** This research received no external funding.

**Institutional Review Board Statement:** The study was conducted in accordance with the Declaration of Helsinki.

**Informed Consent Statement:** Informed consent was obtained from all subjects involved in the study. Written informed consent has been obtained from the patient(s) to publish this paper if applicable.

**Data Availability Statement:** Data are free and available on PUBMED© and Google Scholar©.

**Conflicts of Interest:** The authors declare no conflict of interest.

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
