# Peer review of "Filler Migration after Facial Injection—A Narrative Review"

_cosmetics, doi:10.3390/cosmetics10040115_

Round 1

Reviewer 1 Report

Hello colleagues!

Thank you for the opportunity to review your work. It is detailed and useful for the scientific and practical community.

I would suggest a slightly different presentation logic

In the tables where you indicate complications and methods of treatment, I would introduce a systematization by departments of the face, because. now it looks chaotic.

I would also make a separate section on treatment methods, depending on the period of application and prescription of migration.

It would be very good to indicate the most complicated areas of the face, where complications were detected more often than others.

According to the list of references, I have a question about items that are more than 10 years old. From a historical point of view, this is interesting, but the technology for manufacturing fillers has changed and, perhaps, the mechanism of complications has changed. Please take this into account

Author Response

Reviewer #1

Hello colleagues!

Thank you for the opportunity to review your work. It is detailed and useful for the scientific and practical community.

I would suggest a slightly different presentation logic

In the tables where you indicate complications and methods of treatment, I would introduce a systematization by departments of the face, because. now it looks chaotic.

Response: Thank you, we followed your advice and structured table 2 according to anatomical areas of the face.

I would also make a separate section on treatment methods, depending on the period of application and prescription of migration.

This section has been added. Thank you for suggesting it.

It would be very good to indicate the most complicated areas of the face, where complications were detected more often than others.

This is mentioned now.

According to the list of references, I have a question about items that are more than 10 years old. From a historical point of view, this is interesting, but the technology for manufacturing fillers has changed and, perhaps, the mechanism of complications has changed. Please take this into account

You are right, but patients might have had their injection > 10 years ago.

Reviewer 2 Report

Dear Aurhors,

Congratulations for the manuscript

Best regards

Author Response

Reviewer #2

Dear Authors,

Congratulations for the manuscript

Best regards

Response: Thanks a lot!

Reviewer 3 Report

The authors performed a narrative review of filler migration in patients undergoing facial rejuvenation/volumizing.

After minor revisions, this review will be suitable for publication:

The introduction is too long. Remove 1-2 paragraphs, or move them to the discussion section

Methods:

Please include PRISMA flowchart for your literature search

What was the criteria to qualify as filler migration?

How many millimetres from the site of injection? Is there an established definition of migration? Is 1mm migration significant? 2mm? 5mm? etc

How did the evaluators measure the distance?

How did they know exactly (16 years later, 60 years later, etc.) where the injection point was?

Requires moderate editing, several grammar mistakes

Author Response

Reviewer #3

The authors performed a narrative review of filler migration in patients undergoing facial rejuvenation/volumizing.

After minor revisions, this review will be suitable for publication:

The introduction is too long. Remove 1-2 paragraphs, or move them to the discussion section

Response: We have shortened the introduction.

Methods:

Please include PRISMA flowchart for your literature search

Flowchart was added.

What was the criteria to qualify as filler migration?

How many millimetres from the site of injection? Is there an established definition of migration? Is 1mm migration significant? 2mm? 5mm? etc

This is a very useful question, and the answer is challenging. Thanks for this important remark! Indeed, there is no exact definition, but we selected those articles with a distance > 2 cm to the injection point.

How did the evaluators measure the distance?

The quality of information variable. Some provided cm or mm others not. But an injection in the forehead that results in nodules on the nasal radix or an injection in the zygomatic area producing a nodule on the eyelid have been considered as true cases of migration.

How did they know exactly (16 years later, 60 years later, etc.) where the injection point was?

Again, a crucial point! My guess is that most authors depend on medical history told by the patient. Only a few had a written documentation of used products either. But we don’t have better validated data on this subject.

Reviewer 4 Report

Thank you for your interesting review of filler migration post injection.

It may be more useful to separate Table 2 into permanent and non-permanent fillers, however as there are a number of cases where multiple fillers were used, I can see that this is not easy to do.

Was there any filler in particular that was associated with increased migration?

Where there is diffuse migration of filler, rather than a discrete mass, how do you manage this?

Author Response

Reviewer #4

Thank you for your interesting review of filler migration post injection.

It may be more useful to separate Table 2 into permanent and non-permanent fillers, however as there are a number of cases where multiple fillers were used, I can see that this is not easy to do.

Response: Thank you for this suggestion. We made another attempt in the revised version: patients were separated according to the primary area of injection. However, there are patients with multiple areas been treated.

Was there any filler in particular that was associated with increased migration?

Surprisingly we could not find data on silicone, although it might be expected, that silicone tends to migration since not a tight collagen network is seen around the droplets in contrast to other permanent fillers. The highest percentage of filler migration was reported for polyacrylamide filler.

Where there is diffuse migration of filler, rather than a discrete mass, how do you manage this?

This is a crucial question! Systemic migration has been reported for silicone and PMMA. There is no standardized treatment available. The systemic adverse events bear the risk of mortality. These patients need an interdisciplinary approach. We remember one female with a potentially deadly hypercalcemia.